# Ground Contact Time Estimating Wearable Sensor to Measure Spatio-Temporal Aspects of Gait

**DOI:** 10.3390/s22093132

**Published:** 2022-04-20

**Authors:** Severin Bernhart, Stefan Kranzinger, Alexander Berger, Gerfried Peternell

**Affiliations:** 1Salzburg Research Forschungsgesellschaft mbH, Jakob-Haringer-Straße 5/3, 5020 Salzburg, Austria; stefan.kranzinger@salzburgresearch.at; 2Saphenus Medical Technology GmbH, Magnesitstraße 1, 3500 Krems, Austria; alexander.berger@saphenus.com; 3Ludwig Boltzmann Institut für Experimentelle und Klinische Traumatologie, Donaueschingenstraße 13, 1200 Wien, Austria; gpeternell@trauma.lbg.ac.at

**Keywords:** algorithm design and analysis, gait recognition, medical diagnosis, motion estimation, wearable sensors

## Abstract

Inpatient gait analysis is an essential part of rehabilitation for foot amputees and includes the ground contact time (GCT) difference of both legs as an essential component. Doctors communicate improvement advice to patients regarding their gait pattern based on a few steps taken at the doctor’s visit. A wearable sensor system, called Suralis, consisting of an inertial measurement unit (IMU) and a pressure measuring sock, including algorithms calculating GCT, is presented. Two data acquisitions were conducted to implement and validate initial contact (IC) and toe-off (TO) event detection algorithms as the basis for the GCT difference determination for able-bodied and prosthesis wearers. The results of the algorithms show a median GCT error of −51.7 ms (IMU) and 14.7 ms (sensor sock) compared to the ground truth and thus represent a suitable possibility for wearable gait analysis. The wearable system presented, therefore, enables a continuous feedback system for patients and, above all, a remote diagnosis of spatio-temporal aspects of gait behaviour based on reliable data collected in everyday life.

## 1. Introduction

The appearance of the human gait pattern varies greatly from person to person, due to physical conditions, life circumstances and behaviour, as well as the development of individual techniques based on cognitive and motor skills [1]. For a harmonious and flowing [2] i.e., physiological gait pattern, symmetry (left/right) and dynamics are essential categories. Kuo et al. [3] note that pathologies such as amputations alter a person’s biomechanical properties and refer to studies revealing that atypical walking requires up to twice as much metabolic energy as a typical gait [4,5]. The analysis of unhealthy gait patterns is therefore of great interest in order to be able to develop measures to improve the quality of life of those affected. Sagawa et al. [6] analysed the most relevant biomechanical and physiological parameters for gait analysis of lower-limb amputees in a systematic review. In total, the authors presented 32 parameters and stated that there is no consensus for assessing gait quality due to the variety of parameters used in the literature [6]. However, the results of a previously submitted but not yet published paper reveal that the stance time is a decisive measure for analysing gait quality in foot amputees [7]. Schmid-Zalaudek et al. [7] demonstrate in a study with 865 patients with hip and knee disarticulations, transfemoral, transtibial and foot amputations and 216 able-bodied participants that the most crucial value to reflect symmetry is the stance phase difference between the left and right foot. Following these results, we decided to use ground contact time (GCT) as a representative measurement approach depicting the duration of the stance phase to measure aspects of gait in this study.

According to a systematic review by Chen et al. [8], quantification of gait analysis can be done with conventional, wearable or portable systems. The authors note that some portable systems use electromyography (EMG) for data collection. However, the data interpretability is limited because of large intersubject variability in EMG patterns [9]. Furthermore, Chen et al. [10] mention that conventional approaches such as optical motion capture systems or force plates have high accuracy and precision in analysing gait quality but provide limited continuous monitoring. In contrast, wearable systems such as inertial sensors or sole pressure sensors allow continuous monitoring and data collection and are user friendly for subjects to wear. This advantage opens up broad and diverse possibilities for researchers and practitioners, such as treating physicians, as data can be collected under real-time conditions of the subjects. This could benefit the development of individualized treatments and provide new insights into everyday gait patterns over a long period of time. However, Chen et al. [8] note that the accuracy and precision of such systems depend on the quality of the algorithms provided.

Therefore, the aim of this study is to present an algorithm for wearables to enable continuous and accurate gait analysis. We adopt the measure of Schmid-Zalaudek et al. [7] and choose the difference in GCT between the two feet to evaluate the gait quality of leg amputees. We apply the algorithm developed to data collected from inertial measurement units (IMUs) and a sensor sock, and compare the results with ground reaction force data as ground truth. Extracting GCT difference by subtracting a kinetic and kinematic signal is an innovative method to measure spatio-temporal aspects of gait.

There are several methods for how aspects of gait are measured. For example, the Extra-Laboratory Gait Assessment Method (ELGAM) [11] proposes a method for gait analysis directly at the patient’s home and includes the following parameters to assess gait quality: step length, walking speed, initial walking style, ability to turn the head while walking and static balance. Image processing, floor sensors and wearable sensors are the main measurement methods to use for evaluating gait diagnosis tests [12]. The GAITRite walkways are validated measurement systems to extract spatio-temporal gait parameters based on an integrated ground reaction force data grid [13]. Steinert et al. [14] mention camera-based skeletal tracking with the Microsoft Kinect [15] as an alternative for gait diagnosis when whole-body related parameters are relevant that cannot be captured by ground sensors. Nevertheless, the Microsoft Kinect Sensor provides limited validity for complex gait parameters [15]. Wearable sensors are applied to estimate the gait by inertial sensors, pressure sensors, goniometers or EMG sensors in non-laboratory settings [12]. XSens [16] is a full-body motion capture system that is used for clinical gait analysis and gait parameters extraction based on kinematic movements. However, pedobarographic systems, e.g., Megascan [17], can measure the pressure distribution in the foot by wearable pressure sensing insoles [18]. Experts refer to the isobar pressure distribution on the foot while walking or standing to determine pressure spikes and incorrect loading [19]. Prasanth et al. [20] note that systems combining insole pressure sensors and an IMU, e.g., Moticon [21] and Stridalyzer [22], are emerging in wearable sensor applications to minimize gait recognition errors. Roth et al. [23] present an innovative wearable GCT estimating sensor system combining two pressure measuring insoles with an IMU integrated into each insole. GCT features are extracted by Roth et al. [23] by fusing the sensor sources to increase accuracy metrics. Suralis, the system proposed in this study, differs by a different position of the IMU and a more flexible FSR sensor sock. In addition, only one pressure sensing sock and one IMU are required for the GCT difference computation.

GCT measurement is of great importance in the field of sports and rehabilitation research and makes it possible to derive new insights for athletic coaches and movement therapists. According to Schmidt et al. [24], contact mats and optoelectronic systems provide precise results for recording step parameters, but their applicability is limited to certain spatial conditions. GCT computations by pedobarographic systems are well established by deriving stance and swing phases by integrating the pressure distributions and determining initial contact (IC) and toe-off (TO) events at the signal on- and offsets [25]. However, there are alternative procedures for the calculation of GCT by means of IMUs.

Generally, motion sensors are placed near the motion primitive to be measured. Therefore, Schmidt et al. [24] present a method for measuring GCT by ankle mounted IMUs. The authors derive gait events from the vertical acceleration and gyroscope data based on a study with twelve sprinting able-bodied participants. They report an average deviation of −2.5 ms between the gold standard and their presented method. However, higher accuracies (average deviation between 0 and 2 ms) are reported by Purcell et al. [26] by demonstrating a GCT measurement method during running using shin-mounted accelerometers. The authors developed an algorithm using *x*- and *z*-axis acceleration for extracting the gait events based on a dataset of six healthy runners. McCamley et al. [27] present a method for measuring GCT from vertical acceleration based on a hip-mounted IMU. They stated that stance phases can be achieved from the left and right foot by a single IMU. The developed algorithm and evaluation based on a dataset of 18 able-bodied participants resulted in an average deviation of 17 ms from the gold standard.

Abhayasinghe and Murray [28] present a method for stance phase estimation based on the fusion of acceleration and angular velocity from an IMU put vertically into the trouser pocket. They did not report any accuracy performance metrics but stated that the stance phase can be measured by their sensor setup. Piriyakulkit and Hirata [29] developed an algorithm based on a pressure sensing shoe sole and a thigh mounted IMU at the same leg. They report that stance phases are achievable by this sensor setup based on a dataset of five participants during walking, but the authors did not mention any temporal accuracy performance metrics. The GCT determination algorithms developed within this work are similar to the research of Piriyakulkit and Hirata [29]. However, the algorithms differ in that the IMU is mounted at the thigh of the opposite leg. Therefore, the stance phases of both feet are calculated and the GCT difference can be obtained. In addition, this work includes temporal accuracy performance metrics regarding event detection. Consequently, we want to answer the following research question: Do the algorithms provided enable the analysis of gait quality with regard to the measurement of ground contact time in prosthesis wearers?

## 2. Materials and Methods

### 2.1. Suralis System

For the data acquisition, a modified version of the SAPHENUS Suralis feedback system (Suralis; SAPHENUS Medical Technology, Krems, Austria) was developed (Figure 1). Originally, the Suralis system helps leg amputees in rehabilitation by actuating cuff integrated vibrating motors triggered by sock integrated pressure sensors to feel their missing foot. Therefore, the Suralis system contains a sock equipped with five FSR sensors, each measuring 2.5 × 4.0 cm, for measuring the pedobarography that were located at the heel (heel), metatarsus (met1 and met5) and toes (toes and hallux) (Figure 2). In addition, an IMU (BNO055; Bosch Sensortec, Reutlingen, Germany) for measuring three-dimensional acceleration and angular velocity is available in the Suralis controller unit. For this particular study, the feedback components were removed from the system and a data logging was implemented. Therefore, the custom Suralis system only contains a data logger unit and the original IMU and sensor sock. It is important to note that the FSR sensors do not output units. Therefore, we have not given values on the *y*-axis in Figure 3 and 8, as these would only show the raw value of the ADC input to which the FSR pressure sensor is connected, as neither the sensor nor the voltage divider behave linearly.

To synchronize the data, the sensor sock and the IMU are wirelessly connected via Gazell (Nordic Semiconductor, Oslo, Norway) to a data logger unit, which is connected to a PC via a USB serial interface. The data is then streamed with a sampling rate of 100 Hz and stored in a log file on the PC. The relatively low sampling rate is due to the communication protocol between sensors and the logger. The protocol specifies the maximum payload per transmission and, through the clocking, the maximum transmission frequency. A higher frequency is not possible because we transmit from three devices and need space for possible retransmissions. The sensor sock, IMU and data logger unit are each processed by a microcontroller unit (NRF52840; Nordic Semiconductor, Oslo, Norway). For prosthesis wearers, the sensor sock is worn on the side of the prosthesis (twice on the left, once on the right), while for non-prosthesis wearers it was always placed on the left foot. The IMU is always put into the trousers pocket of the opposite leg (Figure 2).

### 2.2. Experiments

Two data collections were performed. The first dataset was recorded for algorithm development. The dataset contains twelve gait sequences, which include 10 to 20 steps each of straight walking on asphalted level ground. The gait sequences were recorded from three able-bodied adults who had an average weight of 72 kg (sd = 5.7), a height of 176 cm (sd = 8.3) and an age of 28 years (sd = 1.4). Each participant was supposed to conclude four different gaits (limp left, limp right, slow walking, fast walking) to simulate various gait patterns. The limping gait patterns were simulated by short stance phases with reduced load and decreased foot roll-off on the corresponding leg and prolonged stance phases with increased load on the opposite leg. Therefore, we were able to develop and evaluate the algorithm for regular gait patterns with lower and higher stride frequencies (slow and fast walking, respectively) and irregular gait patterns including compensation movements (limping). The participants were equipped with a Suralis and each walk was filmed with a video camera for reference.

The second data recording was conducted for algorithm validation. For the experiment, we collected data from three able-bodied and three prosthesis wearers equipped with the Suralis system. The prosthesis wearers had an average weight of 78 kg (sd = 10.7), a height of 177 cm (sd = 3.7) and an age of 45 years (sd = 14), while the able-bodied persons had an average weight of 77 kg (sd = 4.2), a height of 176.7 cm (sd = 4.5) and an age of 30.3 years (sd = 1.7). Four gait sequences each were acquired with their individual usual gait pattern. Each gait sequence is represented by a complete straight crossing of the GAITRite^®^ walkway (GAITRite; CIR Systems Inc., New York, NY, USA) with a length of 4.27 m and about 4 to 8 steps each on level ground. The GAITRite^®^ walkway was used to acquire a reference from a gold standard system for comparison with the Suralis system. Therefore, a referenced dataset of 2 to 4 strides per gait sequence per participant could be recorded for the algorithm validation. Ethical review and approval were waived for this study because there was no risk of harm to the participants. Informed consent was obtained from all participants.

### 2.3. Data Preprocessing

The video files of the algorithm development data collection were annotated with video labelling software developed in-house. Four different labels were necessary to determine the ground truth in the algorithm development data collection:IC_Left_: The moment of initial contact when the foot touches the ground after a swing phase on the left legTO_Left_: The moment of toe off when the foot leaves the ground after a stance phase on the left legIC_Right_: The moment of initial contact when the foot touches the ground after a swing phase on the right legTO_Right_: The moment of toe off when the foot leaves the ground after a stance phase on the right leg

The annotated events were synchronized with the event timestamps of Suralis and the temporal Suralis gait event detection accuracy was calculated. In the validation data acquisition, an averaged reference GCT per gait sequence was computed from the GAITRite data to extract the Suralis GCT error.

### 2.4. Algorithms

In order to carry out the GCT difference calculation, the gait event detection algorithms for the respective sensors had to be developed based on the algorithm development dataset first. All computations were implemented in R.

Figure 3 shows the signals of the individual FSR sensors of the sensor sock of a stance phase on the left leg. To detect the gait events, all sensor sock channels are summed up first. Afterwards, IC_Sock_ events are detected at the summed signal onsets (increase from 0 to a positive value) and TO_Sock_ events are determined at the summed signal offsets (decrease from a positive value to 0). Occasionally, short signal edges of the heel sensor occur in the signal before a stance phase. To prevent these signal edges from being falsely detected as a stance phase, a time window of 100 ms is checked after each detected event to investigate if it corresponds to the expected signal, i.e., the summed signal remains at a positive value for IC_Sock_ events and remains 0 for TO_Sock_ events in the subsequent time window. The events are discarded by the algorithm if the conditions are not fulfilled.

**Figure 3 sensors-22-03132-f003:**
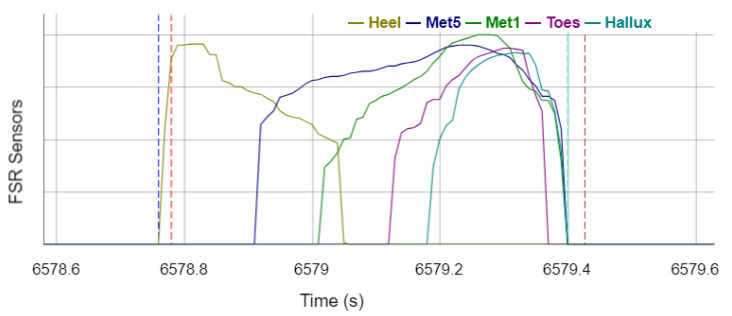
Signals of the FSR sensors of the sensor sock during the stance phase with annotated reference gait events and events recognized by the algorithm. Vertical dotted lines legend: red —reference IC_Left_; brown—reference TO_Left_; blue—sensor IC_sock_; turquoise—sensor TO_sock_.

To detect the gait events at the IMU, the resultant is first calculated from the *X*, *Y* and *Z* acceleration axes, whereby the axes are not fixed to global coordinates because the orientation of the IMU should not matter to ensure easy applicability of the system to non-experienced users: (1)$${\mathrm{Acc}}_{\mathrm{Resultant}}=\sqrt{{\mathrm{Acc}}_{X}^{2}+{\mathrm{Acc}}_{Y}^{2}+{\mathrm{Acc}}_{Z}^{2}}$$

A zero-lag second-order low-pass Butterworth filter with a heuristically determined cut-off frequency of 10 Hz is applied to the acceleration resultant signal to smooth the signal. Figure 4 shows the acceleration resultant and its filtered signal of a stance phase of the right leg. In order to detect the IC_IMU_ events, the smoothed signal is searched for local maxima, whereby a local maximum must be at least an acceleration of 10 m/s² and there must be a distance of 1000 ms (except for the gait type fast walking, where 700 ms applies) to the next local maximum. If a local maximum is detected, it is stored as an IC_IMU_ event and a TO_IMU_ event is searched for in a subsequent time window of 1200 ms (except for the gait type fast walking, where 800 ms applies). In the time window, local maxima are searched for, which again must have an acceleration of at least 10 m/s² and a distance of at least 50 ms between the maxima. Afterwards, a minimum is searched for in a time window of 200 ms before the last detected maximum. This minimum is finally determined as TO_IMU_ event. In this context, local maxima are detected if the current value is larger than the previous and subsequent value, whereas local minima are detected if the current value is lower than the previous and subsequent value in the filtered signal, respectively.

Subsequently, the GCT difference calculation is accomplished via the generated events. First, the GCTs are calculated for each gait sequence (i.e., per gait and per subject) per side (left and right) and per step. Therefore, each gait sequence is first considered individually and for each IC event, a subsequent TO event of the corresponding side is searched. If a corresponding TO event occurs, the GCT is calculated from the temporal difference between the IC and TO events. Thus, the mean GCT per gait sequence and per side can be calculated by averaging the step GCTs. Ultimately, the difference between the mean GCT of both sides results in the GCT difference.

### 2.5. Data Post Processing

To prevent outliers from distorting the results, events that were clearly detected incorrectly are excluded. In the algorithm development data collection, an event is excluded if it deviates more than a heuristically determined threshold of 300 ms from a suitably annotated reference event. No outliers were detected in the sensor sock data, but, the annotated TO reference events in the gait sequence Subject 1—limp right deviate significantly from the other datasets, which could probably be remedied by the reannotation of the reference video. In addition, the gait sequences Subject 1—limp right and Subject 3—limp right are excluded from the analysis because the detection of the gait events in the IMU signal failed for these datasets. The reason for this is probably that the signals of these two datasets differ significantly from the others, as the TO events have the highest peak in both datasets and are therefore incorrectly determined by the algorithm as IC events. Another possible reason is that the limp was different for each subject, as it was only simulated due to the requirements of the experiment.

Gait events of the stance phases at the beginning and end of the recordings were excluded if the detected events deviated significantly from the reference. The reason for this is that the first and last steps of a gait sequence often deviate from the natural gait pattern.

Due to the different stride frequencies of the subjects in the validation dataset, the time window for detecting the TO event in the IMU signal was manually adjusted for the subjects (900 ms for participants P01, P04, P05 and P06; 1000 ms for participants P02 and P03). The algorithms were run on the validation data with the time windows adapted to the walking cadence of the participants. Events that were incorrectly recognized by the algorithm were removed from the generated events by an automated outlier detection with the following implications:Trailing IC implies removing the IC.IC, IC, TO, TO sequence implies removing the first IC and the first TO.IC, IC, TO, IC, TO, TO sequence implies removing the second IC and the second last TO.Leading or trailing stance phases shorter than 500 ms imply removing corresponding IC and TO.

Finally, the GCT difference calculation algorithm was run on the cleaned events.

## 3. Results

### 3.1. Event Detection

Figure 5 shows the mean error of the IC and TO events over all steps per test person and per gait type. The error results from the difference between the timestamps determined by the algorithms and the ground truth (video labels). Since the video was recorded at 30 frames per second, a human error of ±33 ms must be considered in the event detection results, which is visualized in Figure 5 by the grey shaded area.

The results show that the mean error for IC events is largest for the gait type limp left and smallest for fast walking. In general, the error is slightly larger when the IC is done with the right foot instead of the left. The greatest deviations for the TO events occur for the gait type limp left, while the deviations for the other gait types are minimal. Interestingly, for the TO and the IC of the left foot, the gait type limp right shows smaller deviations than limp left.

### 3.2. Ground Contact Time

Figure 6 shows the GCT errors between the Suralis system wearables (IMU/sensor sock) and the ground truth (GAITRite walkway) for four trials. To calculate the GCT error, the ground truth GCT is subtracted from the Suralis GCT for each sensor (IMU/sensor sock). Thus, the GCT error is calculated as follows: (2)$${\mathrm{error}}_{\mathrm{GCT}}={\mathrm{GCT}}_{\mathrm{SuralisSensor}}-{\mathrm{GCT}}_{\mathrm{GroundTruth}}$$

A positive value of the GCT error represents an overestimation and a negative value an underestimation of the GCT of Suralis. It can be seen that for almost all participants and trials the GCTs were overestimated with the sensor sock and underestimated with the IMU.

Figure 7 shows the mean GCT error between the Suralis system wearables (IMU/sensor sock) and the ground truth for prosthesis wearers and non-prosthesis wearers. When prosthesis and non-prosthesis wearers are considered together, it becomes clear that the GCT is underestimated on average for the IMU (−51.7 ms) and slightly overestimated for the sensor sock (14.7 ms). However, the standard deviation is relatively high for both systems. The result of a Mann–Whitney U test [30,31] shows a statistically significant difference in GCT errors between the sensor sock and the IMU. In addition, the deviations are smaller with the sensor sock than with the IMU.

Furthermore, the IMU underestimates the GCT for prosthesis wearers (−63.5 ms) more than for non-prosthesis wearers (−22.8 ms). However, a Mann–Whitney U test [30,31] shows that this difference is not statistically significant. Looking at the results for the sensor sock, there are no noticeable differences between prosthesis wearers (14.9 ms) and non-prosthesis wearers (14.7 ms).

## 4. Discussion

### 4.1. Algorithm Improvements

The results show that the GCTs are on average underestimated with the IMU and slightly overestimated with the sensor sock. In addition, the GCT errors calculated with the sensor sock are smaller on average than with the IMU. Furthermore, it can be shown that with the IMU, lower GCT errors occur on average for non-prosthesis wearers than for prosthesis wearers. This result can probably be attributed to the fact that the algorithm was implemented based on recorded data from non-prosthesis wearers. The presented algorithm can not reach the temporal accuracies of algorithms reported in the literature based on studies involving able-bodied participants [24,26,27]. Nevertheless, the analysis of the results revealed several possibilities for modifying the algorithm provided to reduce the temporal accuracy of the algorithm.

#### 4.1.1. Sensor Sock

Figure 8 shows that the TO is only detected as soon as none of the FSR sensors outputs deliver a value greater than 0. If the value of the Hallux sensor is ignored, the TO would already be detected as soon as the Toes sensor becomes 0. Thus, the GCT would be estimated lower and the GCT error of the sensor sock algorithm would be reduced.

#### 4.1.2. IMU

Figure 9 shows that there are often three peaks in the IMU signal at the IC (instead of one peak presented by Piriyakulkit and Hirata [29]). This increased number of peaks is probably caused by the loose mounting of the IMU in the trouser pocket that is, however, necessary due to higher user acceptance in everyday use. The presented algorithm recognizes the highest peak as the IC, but presumably, the first, smaller peak is the real IC. If the algorithm is adjusted to already detect the first peak, a longer estimation of the GCT would be the result, which would reduce the GCT error of the IMU algorithm.

The time window size for the detection of the TO in the IMU algorithm, which was adapted individually afterwards in the study, must be implemented adaptively depending on the stride rate of the test person so that the algorithm can generally be applied to a large number of test persons in an automated way. The time window for the TO detection must terminate within the subsequent swing phase (Figure 10) so that the last maximum of the stance phase is identified correctly. Based on the data in Table 1, an equation is extracted to implement a time window adaptive to the participants’ stride rate. The constant value of 97,800 is derived by the mean of the products of the participant’s stride rate and applied time window length in the algorithm: (3)$$\mathrm{time}\_{\mathrm{window}}_{\mathrm{adaptive}}=\frac{97,800}{\mathrm{stride}\_\mathrm{rate}}$$

### 4.2. Limitations

Comparing the temporal accuracy of the gait event and GCT detection to the reported accuracies of previous literature [24,26,27], the presented algorithm is less accurate. However, the Suralis system is characterized by its simple usability because the orientation of the IMU in the trouser pocket is not a concern and the sensory sock is easily applicable to leg prosthesis.

Only GCT representing the stance phase was considered for the evaluation of the proposed algorithm. However, the literature includes a spectrum of biomechanical, physiological and other parameters to asses gait quality [6]. Nevertheless, the most commonly used temporal gait features mentioned by Prasanth et al. [20] can be derived from Suralis and evaluated regarding a symmetric gait pattern. In addition, spatial gait features can be extracted from the IMU and spatio-temporal features can be determined by either the IMU or the sensor sock [20]. Therefore, a double-sided application of Suralis enables a complete symmetric gait analysis of temporal, spatial and spatio-temporal features. Moreover, the pedobarographic data of the sensor sock enables the detection of centres of pressure during the foot roll-off, whereby the maximum pressure load during the heel-strike and push-off can be extracted from the FSR sensor signals by creating a cumulative M-shaped curve. Especially, the push-off ground reaction force is a decisive feature in gait analysis of prosthesis wearer because reduced prosthesis-side push-offs cause asymmetric stance phases and thus an irregular gait pattern [32]. However, the computation of reliable ground reaction force features requires a quantitative analysis of Suralis to validate the analogue signals with physical pressure load units. In addition, the Suralis must contain a calibration function to guarantee reliable pressure load data for users with different anthropometrics and body weights.

This study collected data from only three leg amputees with different forms of leg amputation and thus different prosthetic systems. Therefore, in future research, we want to test the provided algorithms on a larger dataset to check their validity regarding different prosthetic systems. In order to validate the proposed algorithm in the context of limping, data from limping individuals should also be collected in the future. This would make it possible to detect gait events in people with a limp, which has not yet worked sufficiently in the context of this study. In addition, Suralis can be adopted for being applied in the gait analysis not only for prosthesis wearers but also for polyneuropathy or paraparesis patients with disabled sense in their feet. In future, the proposed sensor system enables a gait analysis of steps collected in everyday life situations of the patients, which is an immense improvement compared to the stationary clinical gait diagnosis based on only several steps. However, before the proposed system can be applied outside the laboratory, additional knowledge from tests in real-world environments is required.

## 5. Conclusions

This paper presented Suralis, a wearable sensor system device for GCT difference estimation for stance phase symmetry analysis. The GCT difference is achieved by comparing signals gathered by a pedobarographic sensor sock and an IMU put in a trouser pocket at the opposite leg. The algorithm was evaluated on prosthesis wearers as well as able-bodied adults. The results present that the algorithm is a valuable approach with potential for improvements. We can state that the research question posed at the beginning of the study can be answered in the affirmative and that the algorithms provided enable the analysis of gait quality with regard to the measurement of ground contact time in prosthesis wearers. In future, with further development and validation, the proposed sensor system could enable spatio-temporal measurements of gait in everyday environments to assist leg amputees in rehabilitation.

## Figures and Tables

**Figure 1 sensors-22-03132-f001:**
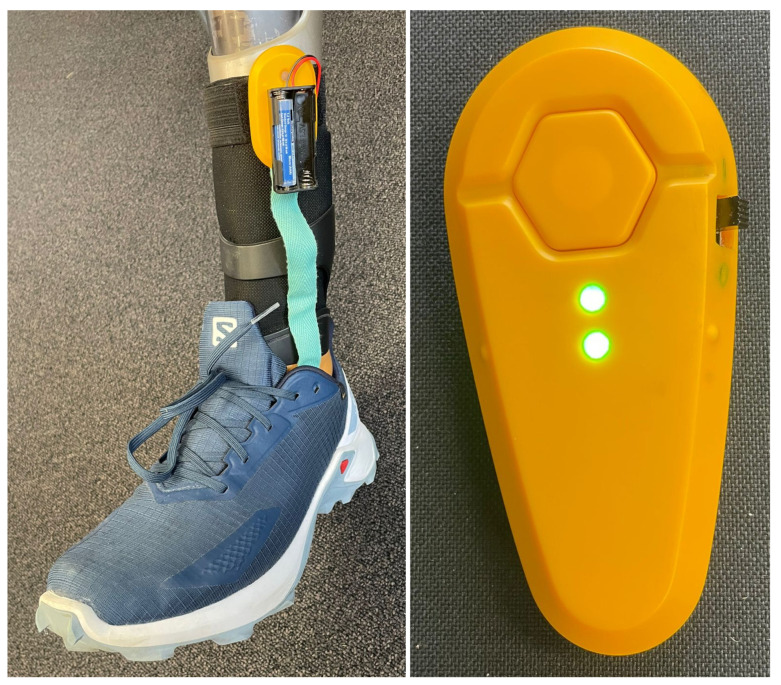
The Suralis sensor systems: The sensor sock worn on a prosthesis within a shoe including the transmitter unit (**left**). The IMU integrated in a housing together with the transmitter unit (**right**).

**Figure 2 sensors-22-03132-f002:**
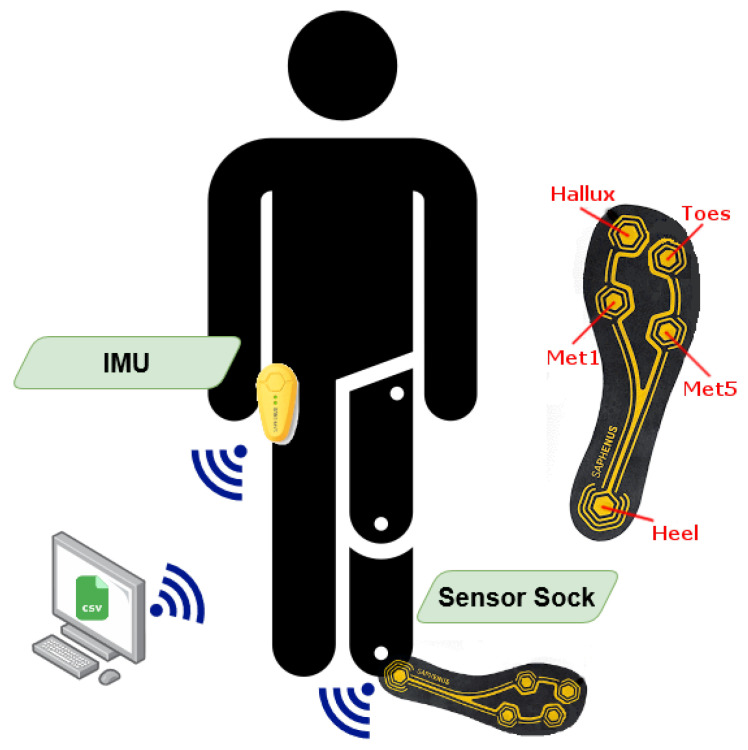
Suralis sensor setup for data acquisition with FSR sensor positions in the sensor sock.

**Figure 4 sensors-22-03132-f004:**
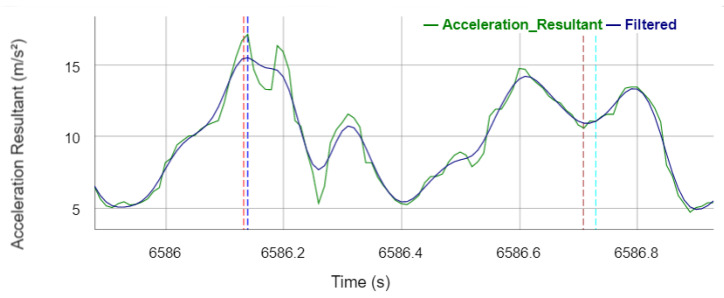
Acceleration resultant and the filtered signal during the stance phase with annotated reference gait events and events detected by the algorithm. Vertical dotted lines legend: red—reference IC_Right_; brown—reference TO_Right_; blue—sensor IC_IMU_; turquoise—sensor TO_IMU_.

**Figure 5 sensors-22-03132-f005:**
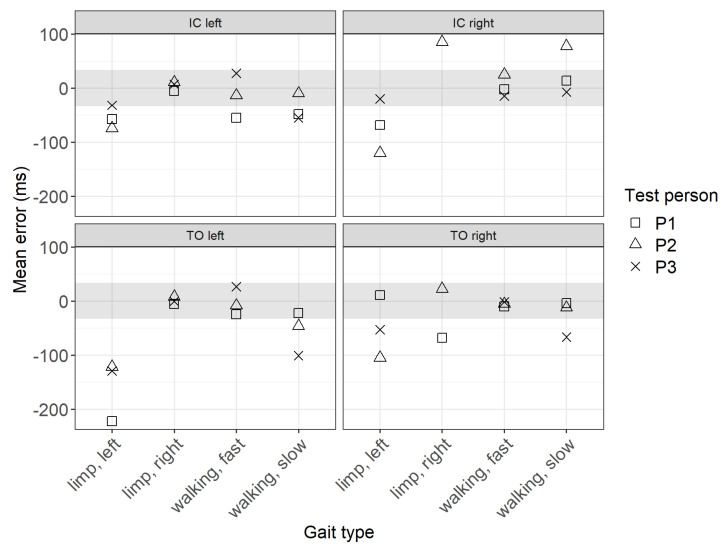
Mean error between the ground truth and sensor data.

**Figure 6 sensors-22-03132-f006:**
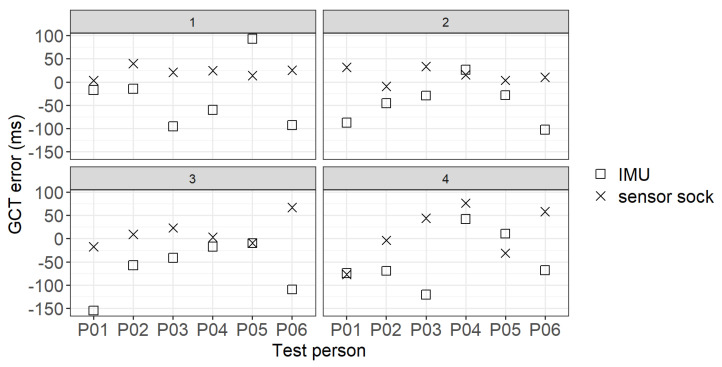
GCT error between the Suralis system wearables (IMU/sensor sock) and the ground truth per participant and trial 1–4.

**Figure 7 sensors-22-03132-f007:**
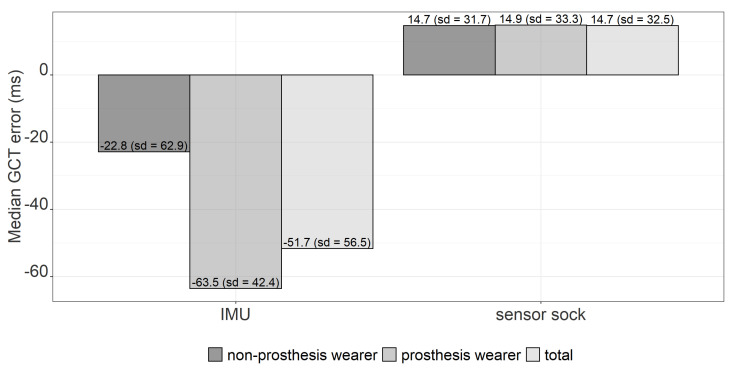
Median GCT error between the Suralis system wearables (IMU/sensor sock) and the ground truth for prosthesis wearers and non-prosthesis wearers.

**Figure 8 sensors-22-03132-f008:**
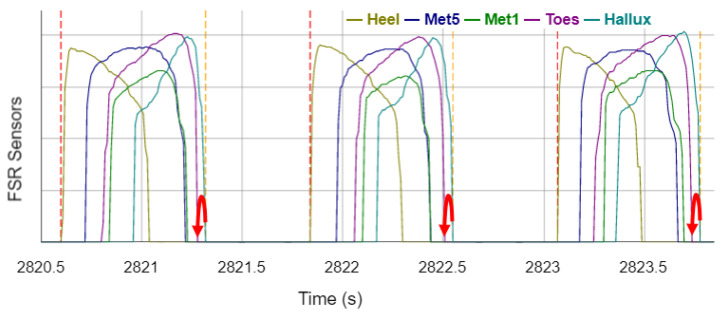
FSR-sensor signals of the sock during three stance phases. Vertical dashed lines visualize the events detected by the algorithm (Vertical dotted lines legend: red: IC_sock_; orange: TO_sock_). Red arrows visualize optimization opportunities.

**Figure 9 sensors-22-03132-f009:**
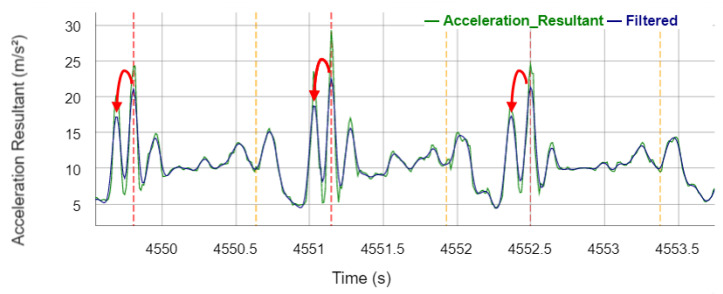
Acceleration resultant signal of the IMU during three stem phases. Vertical dashed lines visualize the events detected by the algorithm (Vertical dotted lines legend: red: IC_IMU_; orange: TO_IMU_). Red arrows illustrate the optimization opportunity.

**Figure 10 sensors-22-03132-f010:**
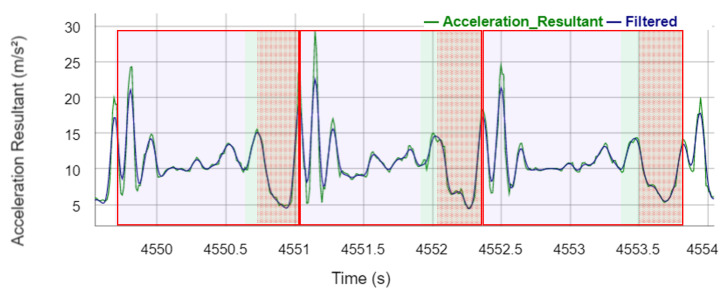
Acceleration resultant signal of the IMU during three steps. Blue shadowed rectangles visualize the stance phase. Green shadowed rectangles illustrate the swing phase. Red transparent rectangles represent the maximum time window for the TO detection. Red shadowed rectangles illustrate the tolerance for the end of the time window.

**Table 1 sensors-22-03132-t001:** Participants’ stride rates (in steps per minute: spm) and their adapted time windows (in milliseconds: ms) for the gait event detection at the IMU in the validation dataset.

Participant	Time Window (ms)	Stride Rate (spm)
P04	900	102.6
P06	900	102.6
P01	900	101.4
P05	900	98.2
P02	1000	95.2
P03	1000	87.8

## Data Availability

The data presented in this study are available on request from the corresponding author.

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
