# Peer review of "Ground Contact Time Estimating Wearable Sensor to Measure Spatio-Temporal Aspects of Gait"

_sensors, 2022, doi:10.3390/s22093132_

Round 1

Reviewer 1 Report

This paper describes the measurement of ground contact time in a system configuration that uses an IMU on one leg and force sensing resister sensors on the other foot.

Although the combination of IMU and pressure sensor is not novel, it is worthwhile that evaluation experiments were conducted on a prosthetic leg wearer.

Opinion

In equation (1), only the absolute magnitude of the accelerometer is used, but it may be better to use a value that takes the vertical direction into account.

For example

  1. Update the upward vector in the interval where 9.81 m/s^2 is detected stably at a suitable time
  2. Use the inner product of the current acceleration and the upward vector as acceleration resultant signal.

If the acceleration resultant signal is in the upward direction, even if not strictly, it would be easier to distinguish the three peaks in Fig. 9 by whether the acceleration is positive or negative.

Minor points

  1. Are there no units on the vertical axis in Fig. 3 and Fig. 8?
    Even if a number such as 200 is written, it is not clear what it indicates.
  2. In the line 56, is "saptio-temporal" "spatio-temporal"?
  3. In the caption of Fig. 1, is "housing togehter"  "housing together"?

Author Response

Dear Reviewer,

thank you very much for your constructive feedback. We agree with most of your comments and we have changed our publication accordingly. In the attached file, we address each of your comments in tabular form and refer to passages in the text that we have changed for applying your feedback to our publication. In addition, all new or changed passages in the manuscript are marked in red.

Kind regards,
Severin Bernhart

Reviewer 2 Report

The appendix is an evaluation of the work.

Author Response

(The authors gave the same response as above.)

Reviewer 3 Report

See attached.

Author Response

(The authors gave the same response as above.)

Round 2

Reviewer 2 Report

The authors addressed all comments and revised the paper. The paper is now much more readable and clear. After reading, I have no further comments.

Author Response

Dear Reviewer,

thank you very much for reviewing our manuscript and your positive feedback.

Kind regards,

Severin Bernhart

Reviewer 3 Report

The changes suggested address my previous concerns with the manuscript, although I do have one remaining suggestion.

As a clinician I find the use of the term "ground contact time" in lieu of stance phase somewhat distracting. It would be useful near the beginning of the paper to include a sentence explaining why the authors prefer the term GCT (presumably because GCT is more descriptive of the measurement approach as opposed to the functional description of a phase of gait), and at some point in the discussion recast the results in terms of stance phase symmetry. This simple addition will make the article more useful to clinicians.

Author Response

Dear Reviewer,

thank you very much for your final feedback. We agree with your comments and added some more information in the manuscript to address your suggestions:

Introduction, lines 33-35: "Following these results, we decided to use ground contact time (GCT) as a representative measurement approach depicting the duration of the stance phase to measure aspects of gait in this study."

Limitations, line 338-339: "Only GCT representing the stance phase was considered for the evaluation of the proposed algorithm. "

Conclusions, line 370-371: "This paper presented Suralis, a wearable sensor system device for GCT difference estimation for stance phase symmetry analysis."

Changed passages are not marked in red as this should be the final submission.

Kind regards,

Severin Bernhart